# Active Learning with Label Comparisons

**Gal Yona**[1,2]      **Shay Moran**[1,3]      **Gal Elidan**[1,4]      **Amir Globerson**[1,5]

[1]Google
[2]Weizmann Institute of Science
[3]Technion
[4]Hebrew University
[5]Tel Aviv University

## Abstract

Supervised learning typically relies on manual annotation of the true labels. When there are many potential classes, searching for the best one can be prohibitive for a human annotator. On the other hand, comparing two candidate labels is often much easier. We focus on this type of pairwise supervision and ask how it can be used effectively in learning, and in particular in active learning. We obtain several insightful results in this context. In principle, finding the best of $k$ labels can be done with $k - 1$ active queries. We show that there is a natural class where this approach is sub-optimal, and that there is a more comparison-efficient active learning scheme. A key element in our analysis is the "label neighborhood graph" of the true distribution, which has an edge between two classes if they share a decision boundary. We also show that in the PAC setting, pairwise comparisons cannot provide improved sample complexity in the worst case. We complement our theoretical results with experiments, clearly demonstrating the effect of the neighborhood graph on sample complexity.

## 1 INTRODUCTION

Supervised learning is a central paradigm in the empirical success of machine learning in general, and deep learning in particular. Despite the recent advances in unsupervised learning, and in particular self-training, large amounts of annotated data are still required in order to achieve high accuracy in many tasks. The main difficulty with supervised learning is, of course, the manual effort needed for annotating examples. Annotation becomes particularly challenging when there are many classes to consider. For example, in a text summarization task, we can ask an annotator to write a summary of the source text, but this will likely not result in the "best" summary. We could also present the annotator a summary and ask for feedback (e.g. is it good), but the quality could be difficult to judge in isolation. We could also ask the annotator to select the best summary out of a set of candidates (e.g. produced by a language model), but this could be taxing if not infeasible when there are many candidates.

Motivated by the above scenario, previous works [e.g., see Stiennon et al., 2020, Ouyang et al., 2022, for a recent application to large language models] have considered an alternative, and arguably natural, form of supervision: "Label Comparisons". Instead of presenting many potential labels to the annotator (e.g., candidate text summaries), we only present two candidates and ask the annotator to choose the better one. For example, when summarizing Snow White, we can ask to compare the summaries "A story about an evil step-mother" and "A story about a girl who is driven to the forest by an evil step-mother and ends up living with dwarves". Most annotators would easily choose the latter as a better summary.

Label comparisons clearly require a much lighter cognitive load than considering all alternatives, and thus have high potential as an annotation mechanism. However, our theoretical understanding of this mechanism is fairly limited. While there has been work on learning to rank, which also uses comparisons, the goal of label comparisons is typically not to learn a complete ranking, but rather to build a model that outputs optimal predictions. Here we set out to analyze label comparisons from this perspective, and we obtain several surprising results and a new algorithm.

Our key question is what is the best way to learn with label comparisons. We assume that during learning we can only ask an annotator for label comparisons and not, for example, for the ground-truth label of the input, which we refer to as an argmax query. We then ask how one can design algorithms that make effective use of such queries, and what is the corresponding query complexity. Namely, how many queries are needed to achieve a given test error. Perhaps the

*Accepted for the 38th Conference on Uncertainty in Artificial Intelligence* (UAI 2022).

most natural way of using comparisons is simply for finding the argmax label, which can be done via $k-1$ active queries. However, as we shall see, this is a suboptimal approach.

The first question we ask is whether access to comparisons is more informative than access to the argmax. If we know all $\binom{k}{2}$ comparisons for $x$, we can also infer the argmax and so it would seem like the answer to this should be in the affirmative. Our first result shows that in the PAC setting, this is in fact not the case, and that knowing all comparisons may result in the same sample complexity as knowing only the argmax. The intuition for this negative result is that for 1D classifiers, the informative points are those that lie close to the decision boundaries between classes, and the argmax label for these points can also be used to find the boundaries, so that comparisons do not provide further advantage.

The negative result above may seem to suggest that comparisons are only useful for inferring the argmax. However, we show that in the case of active learning, comparisons can be used more effectively. We consider the setting where the active learner can choose which label comparison queries to request for a given input $x$ (including not requesting any queries at all). A natural approach here is to take a "standard" active learning algorithm based on argmax queries, and implement it using pairwise comparisons, by using $k-1$ active comparisons for each input $x$ to obtain the argmax. This strategy results in an algorithm that asks $\gamma(k-1)$ comparisons, where $\gamma$ is the number of argmax queries used.

Here we show that one can in fact do better than simulating argmax, by asking the "right" comparisons in an active fashion. These beneficial comparisons are closely related to the "Label Neighborhood Graph" (see Figure 2) where labels are neighbors if they share a decision boundary. We show that it is sufficient to ask queries only about neighbor pairs in this graph. Thus, if this graph is sparse, active learning can be implemented with fewer queries. In particular, for linear classifiers in $\mathbb{R}$, each class has at most two neighbors, and thus the neighborhood graph is very sparse, and our proposed active learning approach is highly effective. Taken together, our results demonstrate the richness of the label-comparison setting, and the ways in which its query complexity depends on the structure of the data.

## 2 RELATED WORK

Several lines of works have addressed alternative modes of supervision for multi-class learning.

**Bandit Feedback:** In this setting (e.g., Kakade et al. [2008], Crammer and Gentile [2013]) the learner only observes whether its predicted class is correct or not. On the one hand, this feedback is stronger than label comparisons, because positive bandit feedback implies knowledge of the argmax. On the other hand, label comparisons provide more information than bandit feedback, because comparisons provide

knowledge about relative ordering of non-argmax labels.

**Maxing from pairwise comparisons.** Maximum selection (maxing) from noisy comparisons is well-studied problem. Falahatgar et al. [2018] give an overview of known results under various noise models. Here, we show that for multi-class learning, using comparisons to first learn the global structure of the problem is more efficient than only using them for maxing. Daskalakis et al. [2011] consider maxing in partially ordered sets, where some pairs may be incomparable, which is interesting to explore in our setting.

**Dueling Bandits:** In online learning, learning from pairwise comparisons is studied under the dueling bandits setting [Saha et al., 2021, Dudík et al., 2015], in which the learner "pulls" a pair of arms and observes the result of a noisy comparison (duel) between them. The objective in these cases is to minimize the regret w.r.t a solution-concept from the social choice literature, such as the Condorcet winner [Yue et al., 2012], Borda winner, Copeland winner, or the Von Neuman winner [Dudík et al., 2015]. The focus on such regret minimization objectives is principally different from ours, since our primary goal is to minimize the number of queries made, rather than minimizing an online loss.

**Active Learning with rich supervision:** Several works have explored alternative forms of supervision. Balcan and Hanneke [2012] explore class-conditional queries, where the annotator is given a target label and a pool of examples, and must say whether one of the examples matches the target label. Several works [Kane et al., 2017, Hopkins et al., 2020, Xu et al., 2017] have studied comparison queries on instances, where the annotator receives two inputs $x_1, x_2$ and reports which one is more positive (for binary classification). Ben-Eliezer et al. [2022] study active learning of polynomial threshold functions in $d = 1$ using derivative queries (e.g., is a patient getting sicker or healthier?). Our supervision is conceptually different from all of these, as it compares between several labels on the same example $x$.

**Learning Ranking as a Reward Signal:** A recent line of work demonstrated that pairwise label-comparisons elicited from humans can be used to improve the performance of LLMs. Stiennon et al. [2020] collect a dataset of human comparisons between summaries of a given text, and use it to obtain better summarization policies, and Ouyang et al. [2022] extend this idea to aligning LLMs with user intent. Our focus here is to understand the theoretical properties of such label comparisons, which we expect will result in more effective ways of collecting and using such comparisons.

## 3 PRELIMINARIES

**Multi-class learning**. Let $\mathcal{X} \subset \mathbb{R}^d$ denote the feature space and $Y$ denote the label space, consisting of $k$ classes. We use $\mathcal{D}$ to denote an (unknown) distribution on $\mathcal{X}$ and $\mathcal{H}$ to denote a class of target functions, $f : \mathcal{X} \to \mathbb{R}^k$. In this

work, our focus is on a realizable setting in which the target function is some (unknown) $f^\star \in \mathcal{H}$. For $\boldsymbol{x} \in \mathbb{R}^d$ and a class $i \in [k]$, $f_i(\boldsymbol{x})$ is the score assigned to class $i$ on instance $\boldsymbol{x}$. Given a target function $f^\star$, the loss of a candidate classifier $f$ is the standard (multiclass) 0-1 loss: $L(f) = \mathbf{Pr}_{\boldsymbol{x} \sim \mathcal{D}}[\arg\max_{i \in [k]} f_i(\boldsymbol{x}) \neq \arg\max_{i \in [k]} f_i^\star(\boldsymbol{x})]$.

Since we are interested in how the difficulty of learning scales with the number of classes $k$, we will explicitly parameterize hypothesis classes in terms of $k$, $\{\mathcal{H}^k\}_{k \in \mathbb{N}}$. For example, the class of homogeneous linear classifiers[1] over $k \in \mathbb{N}$ classes in dimension $d$ is $\mathcal{H}_{\text{lin}}^{k,d} = \{h(\cdot; \boldsymbol{W}) : \boldsymbol{W} \in \mathbb{R}^{k \times d}\}$, where $h(\boldsymbol{x}; \boldsymbol{W}) = \boldsymbol{W}\boldsymbol{x} \in \mathbb{R}^k$.

**Supervision Oracles.** Pertinent to this work is a distinction between two types of access to the target multiclass function: *argmax (i.e. label) queries* and *label-comparison queries*.

**Definition 3.1** (Supervision Oracles). *Given a target function $f^\star : \mathcal{X} \to \mathbb{R}^k$, we define the following oracles:*

$$A^{f^\star}(\boldsymbol{x}) = \arg\max_{i \in [k]} f_i^\star(\boldsymbol{x})$$

$$A^{f^\star}(\boldsymbol{x}, j_1, j_2) = \mathbf{1}[f_{j_1}^\star(\boldsymbol{x}) > f_{j_2}^\star(\boldsymbol{x})]$$

*In the rest of the manuscript we simply use $A^{f^\star}$ to denote the supervision oracle, where it's understood that if it receives an input $\boldsymbol{x}$ it invokes the argmax oracle and if it receives a triplet $\boldsymbol{x}, j_1, j_2$ it invokes the comparisons oracle.*

# 4 PASSIVE LEARNING

We define the sample and query complexities of PAC learnability using both argmax and label-comparisons supervision.[2] We begin with the usual *passive learning* setup, and differentiate between the situation in which every example arrives with its argmax (i.e., the standard PAC setup), and where every example arrives with all the $\binom{k}{2}$ pairwise label comparisons (essentially, the total order on the classes).

**Definition 4.1** (Sample complexity of passive learning with argmax supervision). *Fix a distribution $\mathcal{D}$ over $\mathcal{X}$ and a target function $f^\star : \mathcal{X} \to \mathbb{R}^k$. Let $\mathcal{D}^{f^\star}$ denote the distribution on $\mathcal{X} \times Y$ in which a sample $(\boldsymbol{x}, y) \sim \mathcal{D}^{f^\star}$ is generated by drawing $\boldsymbol{x} \sim \mathcal{D}$ and taking $y = A^{f^\star}(\boldsymbol{x})$.*

*We say that the sample complexity of passively learning a class $\{\mathcal{H}^k\}_{k \in \mathbb{N}}$ is $m_{\mathcal{H}} : (0, 1) \times \mathbb{N} \to N$ if there exists a learning algorithm with the following property: for every distribution $\mathcal{D}$ on $\mathcal{X}$, for every $k \in \mathbb{N}$, for every $f^\star \in H^k$, and for every $\varepsilon \in (0, 1)$, given $m \geq m_{\mathcal{H}}(\varepsilon, k)$ i.i.d samples from $\mathcal{D}^{f^\star}$, the algorithm returns an hypothesis $h$ s.t w.p at least $1 - 1/15$, $L_{\mathcal{D}}(h) \leq \varepsilon$.*

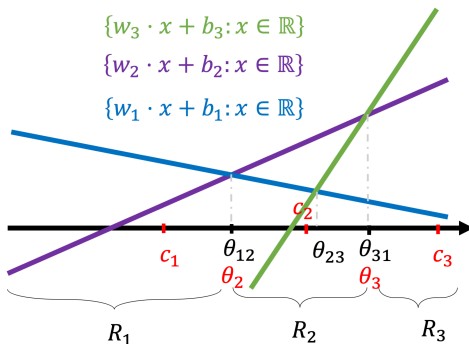

Figure 1: Equivalent view of non-homogeneous linear classifiers in 1d in terms of 1NN classification.

**Definition 4.2** (Sample complexity of passive learning with label-comparisons.). *Fix a distribution $\mathcal{D}$ over $\mathcal{X}$ and a target function $f^\star : \mathcal{X} \to \mathbb{R}^k$. Let $\mathcal{D}^{f^\star}$ denote a distribution on $\mathcal{X} \times \{\pm 1\}^{k^2}$ where a sample $(\boldsymbol{x}, \{b_{ij}\}_{i,j=1}^k)$ is generated by drawing $\boldsymbol{x} \sim \mathcal{D}$ and for $i, j \in [k]$, taking $b_{ij} = A^f(\boldsymbol{x}; i, j)$. We say that the sample complexity of passively learning a class $\{\mathcal{H}^k\}_{k \in \mathbb{N}}$ is $m_{\mathcal{H}} : (0, 1) \times \mathbb{N} \to N$ if there exists a learning algorithm with the following property: for every distribution $\mathcal{D}$ on $\mathcal{X}$, for every $k \in \mathbb{N}$, for every $f^\star \in H^k$, and for every $\varepsilon \in (0, 1)$, given $m \geq m_{\mathcal{H}}(\varepsilon, k)$ i.i.d samples from $\mathcal{D}^{f^\star}$, the algorithm returns an hypothesis $h$ s.t w.p at least $1 - 1/15$, $L_{\mathcal{D}}(h) \leq \varepsilon$.*

Note that in the latter setting, the learner receives strictly more information about every example than in the argmax supervision setting. Namely, the argmax can always be inferred from the total order on the classes. We will therefore consider label-comparisons as helpful in this setup if knowing all comparisons results in improvement to the sample complexity. Our first result is negative: in general, label-comparisons may not be helpful in the passive regime.

**Theorem 4.3.** *Any algorithm that PAC learns $\mathcal{H}_{\text{lin}}^{k,2}$ must use $m_{\mathcal{H}}(\epsilon, k) \in \Omega(k/\epsilon)$ samples, irrespective of whether it has access to argmax or label-comparison supervision.*

*Proof.* For regular PAC learning (with argmax supervision), the standard approach for lower bounding the sample complexity is to lower bound the Natarajan dimension [Natarajan, 1989]. To extend this result to the setting of Definition 4.2, we employ a suitable variant of the dimension introduced in Daniely and Shalev-Shwartz [2014]. Following Brukhim et al. [2022], we refer to it as the Daniely-Shwartz dimension. It provides a tighter lower bound on the sample complexity, and it is also easier to adapt to our label comparison setting.

To emphasize the difference between functions mapping $\boldsymbol{x} \in \mathcal{X}$ to a single class $y \in [k]$ and functions mapping $\boldsymbol{x} \in \mathcal{X}$ to a total order over the $k$ classes, we will denote the former with $f$ and the latter with $f_{\nabla}$ (and likewise for

---

[1]Our convention will be to use homogeneous linear classifiers. Thus when we refer to our results for 1d, we mean the class $\mathcal{H}_{\text{lin}}^{k,2}$.

[2]For simplicity, we consider a PAC notion where the goal is to return $\epsilon$-accurate solutions with constant probability (e.g. 14/15).

hypotheses classes). We write $\arg\max f_\nabla(\boldsymbol{x}) \in [k]$ for the class ranked first in the total order $f_\nabla(\boldsymbol{x})$.

**Definition 4.4.** *Given a set* $\{\boldsymbol{x}_1, \ldots, \boldsymbol{x}_n\} \subset \mathcal{X}$*, we say that* $f_\nabla$ *and* $g_\nabla$ *are* $\boldsymbol{x}_i$*-close if:*

$$\begin{cases} f_\nabla(x_j) = g_\nabla(x_j) & j \neq i \\ \arg\max f_\nabla(x_j) \neq \arg\max g_\nabla(x_j) & j = i \end{cases}$$

With this we can define a variant of Definition 12 in Daniely and Shalev-Shwartz [2014] for the case of extra supervision.

**Definition 4.5** (The Daniely-Shwartz dimension for label comparisons.)**.** *A set* $\{\boldsymbol{x}_1, \ldots, \boldsymbol{x}_n\}$ *is shattered by* $\mathcal{H}_\nabla$ *if there exists a finite subset of functions* $\mathcal{H}'_\nabla \subset \mathcal{H}_\nabla$ *with the following property: for every* $f_\nabla \in \mathcal{H}'_\nabla$ *and for every* $i \in [n]$*, there exists* $g_\nabla \in \mathcal{H}'_\nabla$ *such that* $f_\nabla, g_\nabla$ *are* $\boldsymbol{x}_i$*-close. The Daniely-Shwartz dimension of* $\mathcal{H}_\nabla$*,* $\dim(\mathcal{H}_\nabla)$*, is the maximal cardinality of a shattered set.*

In the Supplementary Material we prove that the sample complexity of passively learning a class $\mathcal{H}$ with label-comparisons (Definition 4.2) is $\Omega(\dim(\mathcal{H})/\varepsilon)$. Thus, our objective is to prove that $\dim(\mathcal{H}_{\text{lin}}^{2,k}) \in \Omega(k)$.

To show this, we will construct a shattered set of size $k$ for $\mathcal{H}_{\text{lin}}^{2,2k}$. Consider $2k$ labels of the form $(b, i)$, where $b \in \{0, 1\}$ and $i \in \{1, \ldots, k\}$. Partition the numbers $1, \ldots, 3k$ to $k$ triples: $\{1, 2, 3\}, \{4, 5, 6\}, \ldots \{3k-2, 3k-1, 3k\}$. We claim that $k$ middle points, $S = \{2, 5, \ldots 3k-1\}$ are shattered by $\mathcal{H}_{\text{lin}}^{2,2k}$. Showing this requires defining a subset $\mathcal{F}$ of $\mathcal{H}_{\text{lin}}^{2,2k}$ with the property of Definition 4.5. To define each function $f_\nabla \in \mathcal{F}$ we will use an equivalent parametrization of linear classifiers in 1d as 1NN classification. i.e., each total order in $\mathcal{H}_{\text{lin}}^{2,2k}$ is parameterized by $\boldsymbol{c} \in \mathbb{R}^{2k}$, where the total order $h(x; \boldsymbol{c})$ is the one implied by sorting the classes according to the distance of their centers $\boldsymbol{c}$ to $x$. See Figure 1 for an illustration. With this parametrization in mind, $\mathcal{F}$ consists of all functions which satisfy the following: for each $i \leq k$, the centers corresponding to labels $(0, i)$ and $(1, i)$ are located in the $i$'th triplet, and *exactly one of them* is located in the middle of the triplet, on the point $3i - 1$. By construction, $|\mathcal{F}| = 4^k$ (for each of the $k$ triplets we need to specify which of the two centers is located in the middle of the triplet, and whether to locate the other center on the left or on the right of it).

To see that $S$ is shattered, consider $f_\nabla \in \mathcal{F}$ and a point $3i - 1 \in S$. W.l.o.g, assume that the center located on $3i - 1$ is $(0, i)$. We define $g_\nabla \in \mathcal{F}$ based on the location of the center of $(1, i)$, which by definition of $\mathcal{F}$, could be either to the right (on $3i$) or to the left (on $3i - 2$). In the first case, $g_\nabla$ is obtained by shifting both centers one unit to the left: in $g_\nabla$ the center $(0, i)$ is located on $3i - 2$ and the center $(1, i)$ is located on $3i - 1$. In the second case, $g_\nabla$ is obtained by shifting both centers one unit to the right: in $g_\nabla$ the center $(0, i)$ is located on $3i$ and the center $(1, i)$ is located on

$3i - 1$. By the definition of $\mathcal{F}$, $g_\nabla \in \mathcal{F}$. Crucially, $f_\nabla$ and $g_\nabla$ are $\{3i - 1\}$-close (Definition 4.4): moving from $f_\nabla$ to $g_\nabla$ the center located on $3i - 1$ (and therefore the argmax) has changed, but the total order for every other point in $S$ is remained unchanged, per the requirement of Definition 4.4. This proves $S$ is shattered, and so $\dim(\mathcal{H}_{\text{lin}}^{2,k}) \in \Omega(k)$, as required. $\qquad\square$

*Remark.* An interesting open question is whether this negative result can be extended to other classes (e.g. linear classifiers in higher dimensions). Technically, this requires lower bounding the the DS dimension of the class, as we did here for $\mathcal{H}_{\text{lin}}^{2,k}$. We conjecture that for $d \gg 1$ the negative result can be extended in a distribution-specific manner (e.g., restricting to distributions with properties such as margin and sparsity); see the discussion in the Supplementary Material, where we report experimental results for the passive learning setting.

# 5 ACTIVE LEARNING

Next, we consider the *active* learning setting. Specifically, we focus on pool-based active learning, where the learner has access to unlabeled samples and can decide which queries to ask the oracle for (including not asking any queries). The performance of the algorithm is now measured in terms of the query complexity, namely the number of queries it makes to the labeling oracle in question.

**Definition 5.1** (Query complexity of active learning.)**.** *The query complexity of actively learning a class* $\{\mathcal{H}^k\}_{k \in \mathbb{N}}$ *is* $q_\mathcal{H} : (0, 1) \times \mathbb{N} \to N$ *if there exists a function* $m_\mathcal{H} : (0, 1) \times \mathbb{N} \to N$ *and a learning algorithm with the following property: for every distribution* $\mathcal{D}$ *on* $\mathcal{X}$*, for every* $k \in \mathbb{N}$*, for every* $f^\star \in \mathcal{H}^k$*, and for every* $\varepsilon \in (0, 1)$*, given* $m \geq m_\mathcal{H}(\varepsilon, k)$ *i.i.d samples from* $\mathcal{D}$ *and at most* $q_\mathcal{H}(\varepsilon, k)$ *queries to* $A^{f^\star}$*, the algorithm returns an hypothesis* $h$ *s.t w.p at least* $7/8$*,* $L_\mathcal{D}(h) \leq \varepsilon$*. We refer to* $q_\mathcal{H}$ *as the query complexity of learning* $\mathcal{H}$ *with argmax supervision or with label-comparison supervision, depending the oracle* $A^{f^\star}$*.*

We note that every active learning algorithm that uses argmax queries can always be simulated using comparison queries: in the adaptive setting (where the choice of query to ask at time $t$ can depend on the previous answers), $k - 1$ label-comparison queries suffice to implement a "tournament" that reveals the argmax. This provides a generic way to use the label-comparison oracle: simply request the label-comparison queries necessary for a "regular" active learner. We therefore say that *comparisons are useful for active learning* if the number of label-comparison queries required to learn a class $\mathcal{H}$ is *strictly lower* than the number of label-comparison queries required to simulate the best active learner that uses argmax queries to learn $\mathcal{H}$.

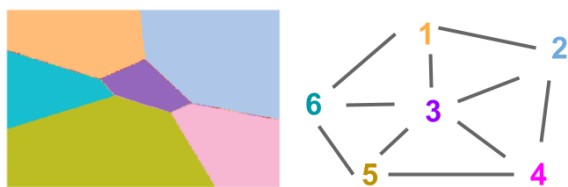

Figure 2: Decision regions of a linear classifier in 2d (left) and its corresponding label neighborhood graph (right).

---

**Algorithm 1** NbrGraphM2B: **active learning of** $\mathcal{H}_{\text{lin}}^{k,d}$ **using** $G$.

> **Input:** $\varepsilon > 0$, a binary active learning algorithm B with query complexity $q_b(\gamma)$, a neighborhood graph $G$.
> **Output:** $f : \mathcal{X} \rightarrow \mathbb{R}^k$.
> **for** $(i,j) \in G$ **do**
>> Use B to learn a binary classifier that distinguishes class $i$ from class $j$ with error at most $\varepsilon/\text{e}(G)$.
> **end for**
> Let $C$ denote the set of all the learned binary classifiers.
> Return $f^{(G,C)}$ (Definition 5.4).

---

Interestingly, the distinction between passive and active learning is important. Our main result is that when the learner is allowed to decide which queries to request, label-comparisons are helpful: we provide a learning algorithm that uses label comparisons more efficiently than simply using them to implement the best "regular" active learner.

**Theorem 5.2.** *The label-comparison query complexity for active learning* $\mathcal{H}_{\text{lin}}^{k,2}$ *is* $\tilde{O}(k \cdot \log \frac{1}{\varepsilon})$*, whereas the query complexity of simulating the best argmax active learner is* $\tilde{\Omega}(k^2 \cdot \log \frac{1}{\varepsilon})$*.*

The proof of Theorem 5.2 will employ a specific multiclass to binary reduction that uses the concept of the *label neighborhood graph* of the target classifier. Intuitively, two classes $i$ and $j$ are considered neighboring if they share a decision boundary; i.e., there are two arbitrarily close points in $\mathbb{R}^d$, where for one the argmax is $i$ and for the other the argmax is $j$. See Figure 2 for an example of the label neighborhood graph of a linear classifier in $d = 2$.

**Definition 5.3** (Label Neighborhood graph). *Fix a continuous function* $f : \mathbb{R}^d \rightarrow \mathbb{R}^k$*. The neighborhood graph* $G = G(f)$ *is an undirected graph on* $k$ *vertices, with an edge between vertices* $i \in [k]$ *and* $j \in [k]$ *if and only if there exists* $\boldsymbol{x} \in \mathbb{R}^d$ *for which for every* $r \in [k]$*,* $f_i(\boldsymbol{x}) = f_j(\boldsymbol{x}) \geq f_r(\boldsymbol{x})$*.*

To simplify notation, we use $(i,j) \in G$ to refer to an edge in $G$, and $\text{e}(G)$ for the total number of edges. The degree of $i \in [k]$ is the number of neighbors $i$ has in $G$.

We next define NbrGraphM2B (Neighborhood Graph Multiclass-to-Binary), a procedure for actively learning a

multiclass classifier $f$ using a neighborhood graph $G$ (see Algorithm 1). Given as input a binary active learning algorithm and a neighborhood graph, it uses comparison queries to learn a binary classifier for distinguishing every pair of neighboring classes $i, j$ in $G$. It then aggregates these into a multi-class classifier using the following scheme:

**Definition 5.4** (Binary to multiclass aggregation.). *Fix* $(G, C)$*, where* $G$ *is a neighborhood graph and* $C = \{h_{ij}\}_{(i,j) \in G, i < j}$ *is a collection of binary classifiers, one for every edge in* $G$*. The graph-based aggregation of* $(G, C)$ *is a function* $f^{(G,C)} : \mathcal{X} \rightarrow \mathbb{R}^k$ *defined as follows:*

$$f_i^{(G,C)}(\boldsymbol{x}) = \frac{\sum_{(i,j) \in G} \mathbf{1}[h_{ij}(\boldsymbol{x}) \geq 0]}{\sum_{(i,j) \in G} \mathbf{1}}$$

*Namely, the label of* $\boldsymbol{x}$ *is the class in* $[k]$ *that won the largest fraction of "duels" against its neighbors in the graph* $G$*.*

An important component in analyzing NbrGraphM2B is the following lemma. It establishes that when invoked w.r.t the true neighborhood graph $G^\star$, if the binary classifiers are sufficiently accurate, then so is the resulting multiclass classifier. See the Supplementary Material for the proof.

**Lemma 5.5.** *Fix a distribution* $\mathcal{D}$ *on* $\mathcal{X}$ *and a classifier* $\boldsymbol{W}^\star$*. Fix* $(G, C)$*. If* $G = G(\boldsymbol{W}^\star)$ *and every* $h_{ij} \in C$ *has error at most* $\varepsilon/e(G)$*, then* $f^{(G,C)}$ *has error at most* $\varepsilon$*.*

From this, we obtain the following upper bound on the query complexity of learning $\mathcal{H}_{\text{lin}}^{k,d}$ using label comparisons.

**Corollary 5.6.** *If the target neighborhood graph* $G^\star$ *is known, the label-comparison query complexity of learning* $\mathcal{H}_{\text{lin}}^{k,d}$ *is* $O(e(G^\star) \cdot q_b(\varepsilon/e(G^\star)))$*, where* $q_b(\gamma)$ *is the query complexity of active learning in the binary case (i.e.* $k = 2$*).*

Corollary 5.6 suggests that label-comparisons will be useful when (i) the target neighborhood graph is sparse (has low degree), and (ii) it can be learned with relatively few label-comparisons. We are now ready to prove Theorem 5.2: we will show that for learning $\mathcal{H}_{\text{lin}}^{k,2}$ (the class for which we demonstrated comparisons are not useful in the passive setting), both these conditions hold. Hence, comparisons indeed provide a provable gain over argmax supervision.

*Proof of Theorem 5.2*: We will begin by instantiating the bound from Corollary 5.6 for $d = 1$. Consider the degree of the neighborhood graph. Using the equivalent parameterization of linear classifiers in $d = 1$ (see Figure 1), it follows that every class $i \in [k]$ has at most 2 neighbors: exactly the preceding and succeeding classes in the sorted order of the classes. Thus, for every $f^\star \in \mathcal{H}_{\text{lin}}^{k,2}$, $\text{e}(G(f^\star)) = O(k)$. Second, active learning in $d = 1$ is well-understood: unlike higher dimensions, the distribution-free query complexity of active learning for two classes is $q_b(\gamma) = \log(\frac{1}{\gamma})$ using binary search over $\mathbb{R}$ [Dasgupta, 2004]. Plugging both of

---

**Algorithm 2** Learning $G(f^\star)$ for $f^\star \in \mathcal{H}_{\text{lin}}^{k,2}$.

---

**Input:** $n$ i.i.d samples from $\mathcal{D}$, $x_1, \ldots, x_n$.
**Output:** A neighborhood graph $G$.

Set $x_L = \min_i x_i$ and $x_R = \max_i x_i$.
Use a comparison sorting procedure to obtain a total order over the $k$ classes, $i_1 \succ \cdots \succ i_k$. Every time the sorting procedure requires the comparison between classes $i, j \in [k]$, determine that $i$ appears before $j$ if and only if (i) $A^f(x_L, i, j) = 1$ and $A^f(x_R, i, j) = 0$, or (ii) $A^f(x_L, i, j) = 1$ and $A^f(x_R, i, j) = 1$.
Define a neighborhood graph $G$ with an edge between $i$ and $j$ iff classes are consecutive in the learned total order.

Return $G$.

---

these facts into the upper bound of Corollary 5.6, we conclude that the query complexity for learning $\mathcal{H}_{\text{lin}}^{k,2}$ when the target neighborhod graph is known is $O(k \cdot \log \frac{k}{\varepsilon})$.

Next, we turn to the question of learning $G^\star$ using label-comparison queries. Towards this, consider Algorithm 2. The algorithm receives a sample of $n = O(1/\varepsilon)$ points from $\mathcal{D}$ and uses exactly $2k \log k$ label comparisons to return a neighborhood graph $G$. As we claim below the graph $G$ will be identical to $G^\star$, except for possibly a set of edges pertaining to classes outside $S$ whose overall probability under $\mathcal{D}$ is smaller than $\varepsilon$. The key observation behind Algorithm 2 is that we can use exactly two label-comparison queries to infer whether a class $i$ appears before a class $j$, as long as both classes are "represented" in $S$.[3] We can therefore use a total of $2k \log k$ queries to infer the total order of all the "represented" classes.

It remains to argue why $O(1/\varepsilon)$ samples suffice to guarantee that with high probability, classes that are not "represented" by $S$ have mass at most $\varepsilon$. To see this, fix $\mathcal{D}$ on $\mathbb{R}$ and denote $F(z) = \mathbf{Pr}_{x \sim \mathcal{D}}[x < z]$. Let $z$ be such that $F(z) = \varepsilon$. We are interested in the number of samples $n$ required to guarantee that $\mathbf{Pr}_{S \sim \mathcal{D}^n}[\min(S) > z] < \delta$. Now,

$$\mathbf{Pr}_S[\min(S) > z] = ((1 - F(z))^n = (1 - \varepsilon)^n \le \exp(-n \cdot \varepsilon)$$

And $\exp(-n \cdot \varepsilon) \le \delta \iff n \ge \frac{1}{\varepsilon} \log \frac{1}{\delta}$. Similarly, the same number of samples can be used to bound the the "tail" beyond $\max(S)$. Union-bounding over both events yields the required result.

To summarize, the full procedure for actively learning $\mathcal{H}_{\text{lin}}^{k,2}$ is to run NbrGraphM2B with the neighborhood graph $G$ that is returned by Algorithm 2. Combining Lemma 5.5 and the analysis of Algorithm 2, we conclude that this procedure has

---

[3]We say a class $i$ is represented in $X$ if the position of $i$ in the total order of all the classes is greater-equal than the position of $\min(S)$ and smaller-equal than the position of $\max(S)$.

---

an overall unlabeled sample complexity of $O(1/\varepsilon)$, and an overall query complexity of $O(k \log k + k \log \frac{k}{\varepsilon}) = \tilde{O}(k \cdot \log \frac{1}{\varepsilon})$.

To conclude the proof of Theorem 5.2, it remains to lower bound the complexity of learning with argmax queries. We will prove that $\Omega(\frac{k}{\log k} \log \frac{k}{\varepsilon})$ argmax queries are needed. This will imply that simulating any argmax active learning requires at least $\tilde{\Omega}(k^2 \cdot \log \frac{1}{\varepsilon})$ label-comparisons. We will prove this via the label revealing task [e.g., see Kane et al., 2017], where the goal is to reveal the correct labels of a given (realizable) sample of $n$ points, and show that $O(\frac{k}{\log k} \log n)$ argmax queries are needed to reveal all $n$ labels.

Towards this, fix $n$ points and consider a tree that denotes the run of an active learning algorithm (with nodes being the queries asked and the children the possible answers). Note that the number of unique labelings corresponds to the number of leaves in the tree and the query complexity corresponds to the depth of the tree, which we denote $q$. The number of ways to arrange $n$ points into $k$ classes in 1d is $k!\binom{n}{k-1}$ ($k!$ options for ordering the classes and then $\binom{n}{k-1}$ options for locating the thresholds). Since the degree of the tree is $k$ for argmax queries, it must be that the $k^q \ge k!\binom{n}{k-1}$, which implies[4] a lower bound $q \ge O(\frac{k}{\log k} \cdot \log n)$.

Together, this concludes the proof of Theorem 5.2. $\square$

Our analysis suggests that when we can efficiently learn $G^\star$ and it is sparse, label-comparisons provide a gain over argmax queries. We showed this when $d = 1$, and it is natural to ask to what happens for $d > 1$. This requires addressing both the question of what is the binary active learning primitive that we use, as well as the questions of sparsity and learning the graph. See the Supplementary Material for a discussion of these aspects.

# 6 A GENERAL PURPOSE ACTIVE LEARNING ALGORITHM

The approach of Algorithm 1 is to explicitly learn $e(G)$ binary classifiers and aggregate them into a single classifier, that is not in $\mathcal{H}_{\text{lin}}^{k,d}$. For simplicity of optimization, we will prefer to work with models in $\mathcal{H}_{\text{lin}}^{k,d}$. To do so, in Algorithm 3 we present the NbrGraphSGD algorithm, a natural variation which can work directly with such models.

It works as follows: we first initialize a multiclass model $h(\cdot; \boldsymbol{W}) : \mathbb{R}^d \to \mathbb{R}^k$ (e.g. $\boldsymbol{W} \in \mathbb{R}^{k,d}$ for a linear model, but $h$ can also be a neural network). For every data point $\boldsymbol{x}$, we sample an edge $(i, j)$ in the graph $G$. This edge is a candidate label comparison. To decide whether to query it or not, we evaluate the difference in logits between labels $i$ and $j$. If this difference is smaller than $\tau$ we query the pair $(i, j)$

---

[4]Using the fact that $\log\left(k!\binom{n}{k-1}\right) = \log k! + \log \binom{n}{k-1} = k \log k + \log\left(\left[\frac{n}{k}\right]^k\right) = k \log k + k(\log n - \log k) = k \log n$

**Algorithm 3** NbrGraphSGD

---

**Input:** Label neighborhood graph $G$, buffer size $R$, steps $T$, confidence parameter $\tau$, learning rate $\eta$, comparison oracle $A^{f^\star}$.

**Output:** classifier $h(\cdot; \boldsymbol{W})$, number of comparisons $q$.

Initialize $\boldsymbol{W}^{(0)}$, $L = 0$, $q = 0$, $b = 0$.

**for** $t = 1, 2, \ldots, T$ **do**

    Sample $\boldsymbol{x} \sim \mathcal{D}$.

    Sample $(i, j)$ uniformly from the edges of $G$.

    **if** $\left| h_i(\boldsymbol{x}; \boldsymbol{W}^{(t-1)}) - h_j(\boldsymbol{x}; \boldsymbol{W}^{(t-1)}) \right| < \tau$ **then**

        Obtain oracle comparison $c = 2(A^{f^\star}(\boldsymbol{x}, i, j) - 0.5)$

        $L \mathrel{+}= \log(1 + e^{-c(h_i(\boldsymbol{x}; \boldsymbol{W}) - h_j(\boldsymbol{x}; \boldsymbol{W}))})$.

        $q \mathrel{+}= 1$, $b \mathrel{+}= 1$.

    **end if**

    **if** $b \geq r$ **then**

        Update $\boldsymbol{W}^{(t)} \leftarrow \boldsymbol{W}^{(t-1)} - \eta \cdot \frac{\partial L}{\partial \boldsymbol{W}}$

        Clear buffer: $L = 0$, $b = 0$.

    **end if**

**end for**

---

and add a binary cross entropy term that encourages the logit difference to have the correct sign. Once we accumulate sufficiently many comparisons, we perform an update step.

The remaining practical question is which graph $G$ to use. Recall that $G^\star$ has an edge $(i, j)$ iff $\exists \boldsymbol{x}$ where $j$ was the 2nd best label and $i$ was the argmax. For 1d, we showed this could be learned from data effectively. We leave the general case open, and consider here practical recipes for $G$. The simplest approach is to base $G$ on prior knowledge regarding which classes are expected to be neighbors (e.g., via distances on their word embeddings, or other co-occurrence statistics). Another practical case is when first and second best labels are available without the $\boldsymbol{x}$ values (e.g., consider asking individuals what are their first and second most favorite products, without keeping user info). Note that in this case, we will receive evidence of edges only for $\boldsymbol{x}$ values sampled from $\mathcal{D}$. This corresponds to an empirical notion of the neighborhood graph, which we define below.

**Definition 6.1** (Empirical Label Neighborhood graph). *For a target function $f^\star : \mathbb{R}^d \to \mathbb{R}^k$, the neighborhood graph $G_{\mathcal{D}}(f)$ is an undirected graph on $k$ vertices, where there is an edge between vertices $i$ and $j$ if and only if there exists $\boldsymbol{x} \in \mathbb{R}^d$ whose probability under $\mathcal{D}$ is non-zero, and for which for every $r \in [k]$, $f^\star(\boldsymbol{x})_i = f^\star(\boldsymbol{x})_j \geq f^\star(\boldsymbol{x})_r$.*

By definition, $G_{\mathcal{D}}^\star \subseteq G^\star$. One might hope that the discarded edges will not impact accuracy w.r.t $\mathcal{D}$. However, in the worst-case this is not true. Specifically, in proving Lemma 5.5 we used the fact that when $B$ is given by the *true* binary classifiers (i.e. $h_{ij} = \boldsymbol{W}^\star{}_i - \boldsymbol{W}^\star{}_j$), the aggregated classifier $f^{(G^\star, C)}$ has perfect accuracy on $\mathcal{D}$. This may fail for

$G_{\mathcal{D}}^\star$: $f^{(G_{\mathcal{D}}^\star, C)}$ may err on examples supported in $\mathcal{D}$; See the Supplementary Material for an example. In Section 7 we observe that the performance of both graphs is comparable.

# 7 EXPERIMENTS

In this section we evaluate our label-comparisons algorithm NbrGraphSGD on synthetic as well as real data.

We consider the online active learning scenario. At each round $t \in [T] = \{1, \ldots, T\}$, the learner receives a batch of points drawn i.i.d. according to $\mathcal{D}$ and must decide which queries to request from the oracle $A^{f^\star}$ (including not requesting any queries). We compare the following methods:

- NbrGraphSGD$(G)$: This is our algorithm which takes as input a graph $G$ and, for a given $\boldsymbol{x}$, only considers label pairs in $G$ as possible pairs to query. For the given $\boldsymbol{x}$, we iterate over all $(i, j) \in G$. For each pair we check if $|W_i \boldsymbol{x} - W_j \boldsymbol{x}|$ is smaller than a fixed threshold. If it is, we query this pair. We consider different versions of NbrGraphSGD$(G)$, that use different graphs.

- PassiveTour: This baseline uses label comparisons to simulate a standard argmax-based active learning algorithm [Joshi et al., 2009]. Namely, for each $\boldsymbol{x}$, we evaluate the logits $W_y \boldsymbol{x}$ and query $\boldsymbol{x}$ if the difference between the first and second best logits is below some threshold. In the standard argmax setting, we would have requested the label of $\boldsymbol{x}$. With label comparisons, we need to do this using $k - 1$ active comparisons. Namely, we perform a tournament between labels to reveal the maximizer.

- ActiveTour: It may seem wasteful to ask for $k - 1$ comparisons as above, since we may be sufficiently confident in some of these comparisons. We thus consider an "active tournament" algorithm: whenever the current model is sufficiently confident in a given pair in the tournament, we take the model's answer, and do not query for it.

**Evaluation.** In online active learning, the quality of an algorithm is measured by its accuracy after $T$ rounds, and the total number of comparisons requested within these $T$ rounds. We use a linear teacher model to simulate the comparison oracle (Definition 3.1), and measure accuracy as the categorical accuracy[5] on the test set, w.r.t the teacher's argmax. We use an *accumulating buffer* mechanism to control for the number of parameter updates across methods (each method accumulates the requested comparisons until the buffer is full, and only then performs a gradient update).

## 7.1 SYNTHETIC DATA

In this section we validate our theoretical findings from Sections 4 and 5 on synthetic data. Specifically, for $d \in \mathbb{N}$

---

[5]Specifically, use Top-K accuracy, where $K = 0.1 \cdot k$.

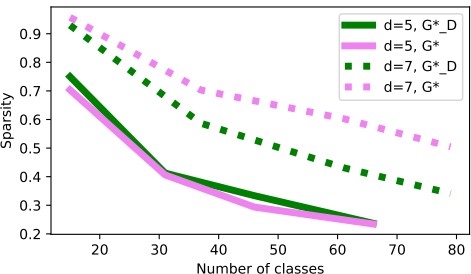

Figure 3: Sparsity level for a random linear model as a function of the number of effective classes $k$ for $d = 5, 7$

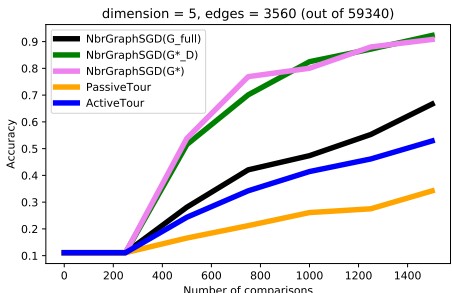

Figure 4: Comparing algorithm `NbrGraphSGD` w.r.t $G_\mathcal{D}^\star$ (green) and $G^\star$ (purple) against three baselines: passive tournament (yellow), active tournament (blue) and algorithm `NbrGraphSGD` with respect to a complete graph (black).

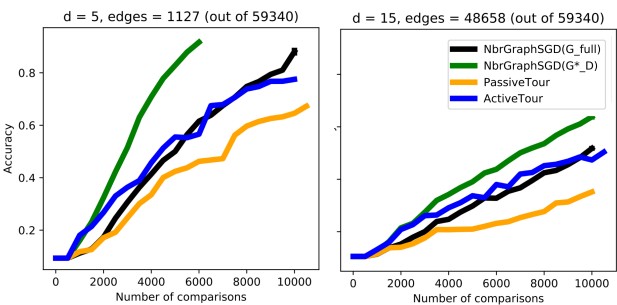

Figure 5: Comparing the performance of `NbrGraphSGD` w.r.t $G_\mathcal{D}^\star$ (green) against the baselines on the QuickDraw dataset. In the plot titles, $d$ denotes the dimension of low-dimensional projection of the data and edges is $e(G^\star)$, the number of edges in the true neighborhood graph of $\boldsymbol{W}^\star$.

and $\hat{k} \in \mathbb{N}$ we consider $\mathcal{D}$ to be the uniform distribution on a unit sphere in $\mathbb{R}^d$ and draw a random linear target model $\boldsymbol{W}^\star \in \mathbb{R}^{\hat{k},d}$. This yields a multiclass classifier with $k \leq \hat{k}$ distinct decision regions ("effective classes"). We draw data from $\mathcal{D}$ and divide it into training and test sets.

**Sparsity of the neighborhood graph.** We begin by computing the sparsity level of both the true neighborhood graph $G^\star = G(\boldsymbol{W}^\star)$ and the empirical neighborhood graph $G_\mathcal{D}^\star = G_\mathcal{D}(\boldsymbol{W}^\star)$, where the latter is computed w.r.t the training set. We define the *sparsity level* as the number of edges in $G$, divided by $\binom{k}{2}$ (i.e,. the number of edges in a complete graph). In Figure 3 we plot the sparsity level as a function of $k$ and $d$, as averaged over 25 random target models. We see that the empirical sparsity level tracks the true sparsity level, and that for a fixed dimension $d$, both decrease with the number of effective classes $k$. This confirms that we expect the sparsity to "kick in" when $k \gg d$.

**Comparisons of Active Learning Methods.** We next compare the different baselines described above. For `NbrGraphSGD`$(G)$ we consider multiple variations, that use different versions of the graph $G$. In Figure 4 we report the performance of `NbrGraphSGD` relative to several natural baselines. First, it can be seen that the active tournament outperforms the passive one, suggesting that indeed some tournament queries can be avoided. Yet `NbrGraphSGD` outperforms the tournament baselines, indicating that tournament comparisons are generally not the optimal approach. Within the `NbrGraphSGD` methods, using the true graph (either $G^\star$ or $G_D^\star$) provides the best performance, indicating that the graph plays an important role in active learning efficacy, and that `NbrGraphSGD` can use this structure.

## 7.2 REAL DATA

The QuickDraw dataset [Ha and Eck, 2017], is a collection of 50 million drawings across 345 categories, contributed by players of the game "Quick, Draw!". We use the bitmap version of the dataset, which contains these drawings converted from vector format (keystrokes) into 28x28 grayscale images. We randomly select $70,000$ examples from this

large data and use $60,000$ as our training set and the rest as the test set. We train a linear teacher on the data after randomly projecting it into $\mathbb{R}^d$. We then use the resulting model $\boldsymbol{W}^\star$ to implement the label-comparison oracle (see Definition 3.1). We denote the true graph of $\boldsymbol{W}^\star$ (Definition 5.3) as $G^\star$ and the empirical graph of $\boldsymbol{W}^\star$ (Definition 6.1, as computed w.r.t the training set) as $G_\mathcal{D}^\star$.

We begin by comparing the performance of `NbrGraphSGD` w.r.t $G_\mathcal{D}^\star$ with the same baselines from Section 7.1. We explore the relationship between the sparsity of the *true graph* $G^\star$ and the performance of `NbrGraphSGD` w.r.t $G_\mathcal{D}^\star$ as a function of the dimension $d$ and $k = 345$. In Figure 5 we report the query complexities w.r.t $d = 5$ (left) and $d = 15$ (right). Note that this is a realizable learning task since the models are measured in terms of their accuracy w.r.t the teacher's predictions, and the teacher is also a linear model. In line with our theoretical results from Section 5, we observe that the gain from using our method (over e.g. the passive or active tournament baselines) is smaller when the true neighborhood graph is less sparse.

## 8  CONCLUSIONS

We studied the setting where annotators are asked to provide only pairwise label comparisons. We believe this is a natural setting as it is both easy for humans to provide, and still results in sufficient information for learning. Our results provide several key characterizations of how this information should be gathered and used. We show that, perhaps counter-intuitively, there are cases for which having all the class comparisons per training point does not yield a sample complexity advantage over just receiving the one true class label. On the other hand, in the active setting, we show that comparisons can be used in an effective way that goes beyond obtaining the argmax training labels.

Many interesting open questions remain. First, our focus was on linear classification, and it would be interesting to generalize the result to other classes (such as neural networks). Second, one can consider a mixture of comparisons and true-labels, since the latter may be easy to obtain in some instances, and hence query-complexity should count these cases differently. Finally, here we assumed that annotators can provide answers to all queries. In practice, some queries may not be answerable (e.g., labels are too "close" or both are equally bad), and it would be interesting to extend the formalism and practical algorithm to these cases.

### Acknowledgements

We thank Ami Wiesel for contributing many ideas throughout the development of this work and for helpful feedback on this manuscript.

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
