# OpenReview forum: "Active Learning with Label Comparisons"
_auai.org/UAI/2022/Conference — UAI 2022 Poster_

### Official Review · Reviewer_qavb · 2022-04-12

**Q2(1) Originality/Novelty:** 2
**Q2(2) Significance/Impact:** 2
**Q2(3) Correctness/Technical Quality:** 3
**Q2(6) Clarity Of Writing:** 3
**Q6 Overall Score:** 6
**Q8 Confidence In Your Score:** 3

**Q1 Summary And Contributions:**

This paper considers the active learning setting where human annotators are provided with label comparisons instead of an overall ground-truth label. Their first result shows that in the PAC setting knowing all ${k \choose 2}$ combinations has the same complexity as knowing the $\argmax$ of the labels. They also prove the sample complexity for the pool-based active learning setting. They propose a new graph-based active algorithm NbrGraphSGD(G) and evaluate it empirically.

**Q2 Assessment Of The Paper:**

More detailed information regarding each of these aspects is given below:

**Q2(4) Quality Of Experiments (Optional):**

2: Fair: The experimental evaluation is weak: important baselines are missing, or the results do not adequately support the main claims.

**Q2(5) Reproducibility:**

3: Good: Key resources (e.g., proofs, code, data) are available and key details (e.g., proofs, experimental setup) are sufficiently well-described for competent researchers to confidently reproduce the main results.

**Q3 Main Strengths:**

1) They show in Theorem 4.3 that in the 1D setting the $\arg\max$ oracle has the same complexity as the label comparison oracle. Similarly, they show in Theorem 5.2 for the active pool-based setting, that the sample complexity of label comparison is actually smaller than the lower bound for the sample complexity for the $\arg\max$ oracle. Again the result only holds for $d=1$.

2) They propose a new graph-based active learning algorithm NbrGraphSGD(G) that uses this label comparison by building neighborhood graphs.

3) They experiment in synthetic and real-dataset against a few benchmarks.

**Q4 Main Weakness:**

1) Their first main result entirely depends on the 1-dimensional case. It is not clear to me what will happen in a large, or sparse dimensional setting. We surely do not hope that even in these large dimensions the most informative samples lie close to the decision boundary. Some comment on this will be helpful.

2) I am not sure that the discussion after Definition 2 for the pool-based setting is true: *Every active learning algorithm that uses argmax queries can always be simulated using comparison queries*. What happens to a core-set based approach (Sener and Savarese, 2017). In that case, choosing samples from your unlabelled pool that maximally covers it using minimum samples and asking for their label comparisons will take far more sample complexity than just choosing samples based on their uncertainty score.

**Q5 Detailed Comments To The Authors:**

See 1), 2) from Q4.

3) What does the $O^{\tilde}$ hide in the result of Theorem 5.2. Also, is Algorithm 2 (used for proving Theorem 5.2) only hold for $d=1$? If yes, then how do you use it for NbrGraphSGD(G)? The NbrGraphSGD requires the neighborhood graph $G$ as input, and do you use Algorithm 2 to construct it?

4) The baselines chosen for the experimental setting seem to be inadequate. For the pool-based setting, a lot of recent advances have been made for active learning algorithms. At least for the real-dataset, I suggest experimenting against BALD, Coreset, Badge, etc. Note that in section 6 you state that $h$ can also be a neural network. So I do not see any reason not to test against Badge?



**Q7 Justification For Your Score:**

Refer to Q3, Q4, Q5. If the authors answer my queries sufficiently in Q4, Q5, I am willing to raise my scores.

**Q9 Complying With Reviewing Instructions:**

1: Yes.

---

### Official Review · Reviewer_V9Y6 · 2022-04-14

**Q2(1) Originality/Novelty:** 2
**Q2(2) Significance/Impact:** 1
**Q2(3) Correctness/Technical Quality:** 3
**Q2(6) Clarity Of Writing:** 2
**Q6 Overall Score:** 4
**Q8 Confidence In Your Score:** 3

**Q1 Summary And Contributions:**

The paper studies an active learning setting where annotators are meant to provide pairwise label comparisons. It is quite reasonable to assume this is cognitively less demanding compared to providing all labels and asking the annotators to provide the argmax.

**Q10 Ethical Concerns (Optional):**

No ethical concerns.

**Q2 Assessment Of The Paper:**

More detailed information regarding each of these aspects is given below:

**Q2(4) Quality Of Experiments (Optional):**

3: Good: The experimental evaluation is adequate, and the results convincingly support the main claims.

**Q2(5) Reproducibility:**

2: Fair: Key resources (e.g., proofs, code, data) are unavailable but key details (e.g., proof sketches, experimental setup) are sufficiently well-described for an expert to confidently reproduce the main results.

**Q3 Main Strengths:**

It’s great to see a combination of theoretical and experimental evaluation and also good that the latter supports the former. Also there are both synthetic and real data and they seem to follow the same trend.

**Q4 Main Weakness:**

My main concern is the actionability of the results and their significance. Basically the takeaway message is that in passive learning setting  there are cases for which having all the class comparisons for a data point does not necessarily yield an advantage over receiving the argmax. On the contrary, in active learning setting in some cases comparisons are more effective than argmax. My main question is the following: I have a classification problem in hand that I require labels for. How do I use the current finding to choose what’s the best way forward? I create a neighbourhood graph and check it’s sparsity? Or there are other guidelines?

**Q5 Detailed Comments To The Authors:**

- The evaluation has a theoretical part and an experimental part. The proofs for technical part are included. There is not much details on the experimental part though. They wouldn’t be reproducible given the level of details currently given.
- The Introduction and related work are quite clearly written, but the rest is very hard to follow. Figures are not necessarily close to their description. Algorithms are referenced in the text as a whole, without a line by line clear cut descriptions. Some proofs include few definitions within them, which makes it very hard to follow.

The work needs to be motivated better. The use cases of the findings need to be outlined better. What happens if labels are required? It would strengthen the message of the paper is some systematic guidelines were generated from the findings that address all applicable scenarios.

**Q7 Justification For Your Score:**

While there are some original ideas, I struggle to see how this can be used in practice. Clarifying that would be a huge help for the significance and a good improvement overall.

**Q9 Complying With Reviewing Instructions:**

1: Yes.

---

### Official Review · Reviewer_nAYf · 2022-04-25

**Q2(1) Originality/Novelty:** 3
**Q2(2) Significance/Impact:** 3
**Q2(3) Correctness/Technical Quality:** 3
**Q2(6) Clarity Of Writing:** 4
**Q6 Overall Score:** 7
**Q8 Confidence In Your Score:** 2

**Q1 Summary And Contributions:**

1. This paper theoretically shows that knowing all label comparisons does not result in improvement to the sample complexity in the passive learning regime.

2. A new active query approach based on label neighborhood graph is proposed for active learning with label comparisons.

3. This paper shows that label-comparison-based query approach can improve the query complexity from both theoretical and empirical perspectives.

**Q10 Ethical Concerns (Optional):**

Not applicable.

**Q2 Assessment Of The Paper:**

More detailed information regarding each of these aspects is given below:

**Q2(4) Quality Of Experiments (Optional):**

3: Good: The experimental evaluation is adequate, and the results convincingly support the main claims.

**Q2(5) Reproducibility:**

3: Good: Key resources (e.g., proofs, code, data) are available and key details (e.g., proofs, experimental setup) are sufficiently well-described for competent researchers to confidently reproduce the main results.

**Q3 Main Strengths:**

1. The problem studied in this paper is very interesting. As the authors mentioned, query from all classes is difficult even impossible when the number of classes is large, yet querying the label comparisons is a very natural alternative strategy in real-world tasks.

2. This paper shows that although learning from label comparisons does not benefit the sample complexity in the passive learning, label-comparison-based query can improve the query complexity in active learning setting from both theoretical and empirical perspectives.

**Q4 Main Weakness:**

1. Empirical analysis on passive learning with label comparisons could be conducted to support the results in Section 4.

2. For active tournament algorithm, a label pair would be selected if the current model is "sufficiently" confident. How does the confidence be measured?

3. In figure 3, it seems that when d=7, the gap between the sparsity levels of empirical neighborhood graph and true neighborhood graph is larger. Figure 4 only shows the results with d=5. What about the accuracy comparison when d=7?

**Q5 Detailed Comments To The Authors:**

This paper considers the active learning setting in which label pairs (instead of max labels) are queried for manual annotation. The authors offer an approach based on label neighborhood graph and shows it improves the query complexity compared with regular active learning from both theoretical and empirical perspectives. The studied problem is very interesting and the proposed approach is technically sound.

As I mentioned in Q4, more empirical analyses, including the performance comparison in passive learning and comparisons with varied d, could be further conducted.


**Q7 Justification For Your Score:**

The authors proposed a new active learning approach based on label comparisons and showed it improves the query complexity compared with regular active learning from both theoretical and empirical perspectives. The query setting studied in this paper is realistic. The proposed approach is technically sound.


**Q9 Complying With Reviewing Instructions:**

1: Yes.

---

### Decision · Program_Chairs · 2022-05-15

**Decision:**

Accept (Poster)

**Comment:**

Meta Review: Generally, the idea of querying the label comparisons to is an interesting and natural choice to enable active learning in real-world tasks. Both theoretical analyses and empirical studies have been reported in this paper. The whole paper is well organized and easy to follow.

The expeirmental studies performed in this paper can be improved, such as considering performance comparison in passive learning, the inclusion of more SOTA baselines, etc.